# Comparative analysis of the *ex vivo* IFN-gamma responses to CD8+ T cell epitopes within allelic forms of P*f*AMA1 in subjects with natural exposure to malaria

**Omarine N. Nlinwe[1][¤], Ebenezer A. Ofori[1], Kwadwo Akyea-Mensah[1], Eric Kyei-Baafour[1], Harini Ganeshan[2,3], Maria Belmonte[2,3], Bjoern Peters[4], Eileen Villasante[2], Martha Sedegah[2], Kwadwo Asamoah Kusi[1] ***

**1** Immunology Department, Noguchi Memorial Institute for Medical Research, College of Health Sciences, University of Ghana, Legon, Accra, Ghana, **2** Malaria Department, Naval Medical Research Center, Silver Spring, Maryland, United States of America, **3** Henry M. Jackson Foundation for the Advancement of Military Medicine, Bethesda, Maryland, United States of America, **4** Infectious Disease and Vaccine Center, La Jolla Institute for Immunology, La Jolla, California, United States of America

¤ Current address: Faculty of Health Sciences, Department of Medical Laboratory Science, The University of Bamenda, Bamenda, Cameroon
* akusi@noguchi.ug.edu.gh

**Data Availability Statement:** All relevant data are within the manuscript and its Supporting information files.

## Abstract

Antigen polymorphisms in essential malarial antigens are a key challenge to the design and development of broadly effective malaria vaccines. The effect of polymorphisms on antibody responses is fairly well studied while much fewer studies have assessed this for T cell responses. This study investigated the effect of allelic polymorphisms in the malarial antigen apical membrane antigen 1 (AMA1) on *ex vivo* T cell-specific IFN-γ responses in subjects with lifelong exposure to malaria. Human leukocyte antigen (HLA) class I-restricted peptides from the 3D7 clone AMA1 were bioinformatically predicted and those with variant amino acid positions used to select corresponding allelic sequences from the 7G8, FVO, FC27 and tm284 parasite strains. A total of 91 AMA1 9-10mer peptides from the five parasite strains were identified, synthesized, grouped into 42 allele sets and used to stimulate PBMCs from seven HLA class 1-typed subjects in IFN-γ ELISpot assays. PBMCs from four of the seven subjects (57%) made positive responses to 18 peptides within 12 allele sets. Fifty percent of the 18 positive peptides were from the 3D7 parasite variant. Amino acid substitutions that were associated with IFN-γ response abrogation were more frequently found at positions 1 and 6 of the tested peptides, but substitutions did not show a clear pattern of association with response abrogation. Thus, while we show some evidence of polymorphisms affecting T cell response induction, other factors including TCR recognition of HLA-peptide complexes may also be at play.

**Funding:** This study was funded by a Noguchi Postdoctoral Training in Infectious Diseases Research program award (Omarine Nlinwe). Grant for this training program was originally from the Bill and Melinda Gates Foundation awarded to the Noguchi Memorial Institute for Medical Research (Global Health Grant number OPP52155, Kwadwo Koram). The funders had no role in study design, data collection and analysis, decision to publish, or preparation of the manuscript.

**Competing interests:** The authors have declared that no competing interests exist.

# Background

Malaria remains an economically important disease globally as it results in a huge loss in productivity amongst affected populations and requires significant financial inputs for control and prevention [1]. Although *Plasmodium vivax* is the most widespread causative agent globally, *Plasmodium falciparum*, which is found mostly in Africa, has overwhelmingly been responsible for the bulk of malaria-related mortality globally [1]. Concerted application of available disease control and prevention tools has resulted in significant decline in malaria-related morbidity and mortality [1], but given the fact that progress in malaria control has plateaued since 2017 [1] the WHO Global Malaria Strategy for 2016 to 2030 therefore clearly articulates the need for additional complementary tools to ensure disease elimination and eradication [2]. The World Health Organization recognizes an effective vaccine as one such complementary tool, but the development of a broadly effective malaria vaccine has been challenged by the complex nature of the malaria parasite and our incomplete understanding of the parasite's interaction with the host's complex immune system. The huge diversity of parasite strains presents the parasite with a significant immune evasion advantage as the infected host's immune system is unable to effectively deal with parasite variants that it has not previously encountered. The most advanced malaria vaccine, RTS,S, is a subunit vaccine that contains portions of the *P. falciparum* sporozoite surface antigen known as circumsporozoite protein (CSP) from the 3D7 parasite strain and has received a positive scientific opinion from the European Medicines Agency [3]. Following phase III trials in Africa, the vaccine showed an efficacy of up to 50% after 12 months [4], and this declined to less than 10% after a 7-year follow up of vaccinated children [5]. This modest short-term efficacy of the RTS,S vaccine has in large part been attributed to an inefficient induction of potent helper T cell responses by the vaccine [4,6]. Cumulative vaccine efficacy in 5–17 month old infants was 50.3% in those with infections that were similar to the vaccine strain, compared to a 33.4% vaccine efficacy in mismatched infections over one year [7]. Similar allele-specific vaccine efficacy has been reported in trials of vaccines containing other polymorphic antigens [8,9] as well as those containing attenuated whole sporozoites [10]. A 5-dose regimen of the PfSPZ Vaccine administered to malaria-naïve adults resulted in 92.3% protection against experimental challenge with sporozoites of the homologous parasite strain and 80% protection against heterologous parasites three weeks post-immunization [10]. Protection was however up to 70% against homologous parasite challenge and only 10% against heterologous parasite re-challenge 24 weeks after the last immunization [10].

There is abundant literature on this specific effect of allelic polymorphisms on the generation of antibody responses [11–13], but relatively little is known about this effect in the generation of T cell responses. T cells respond to antigen peptides packaged with major histocompatibility complex (MHC) molecules using their T cell receptors (TCRs). Both the MHC molecules and TCR interact with the peptide by identifying and binding to specific amino acid anchor residues within the peptide [14,15]. It is therefore possible that drastic changes in these essential amino acid sequences could interfere with MHC recognition of peptides and affect antigen presentation to T cells, or interfere with TCR recognition of MHC presented peptides and hence abrogate T cell activation. In a recent trial of a 3D7 strain of CSP and apical membrane antigen 1 (AMA1) gene-based malaria vaccine (DNA prime followed by boosting with viral vector) in malaria-naïve adults, sterile protection in some vaccinees was associated with their CD8+ T cell recognition of two MHC class 1-restricted immunodominant peptides in the vaccine strain of AMA1. Immune cells from these protected vaccinees, with CD8+ T cell memory responses to the 3D7 strain of AMA1 (vaccine strain), failed to respond to corresponding allelic peptides from the 7G8 strain of

AMA1, even though the two sets of allelic peptides only differed by a single amino acid [16]. Some other studies have described similar effects of allelic polymorphism on T cell responses against *Plasmodium* and other viral antigens [17–21]. We have also recently reviewed the available literature on the effect of antigen polymorphisms on the induction of T cell responses to malarial antigens [22]. Thus, allelic polymorphism may significantly influence the induction of T cell responses against MHC-restricted peptides from parasite antigens which are critical for protective immunity.

This study therefore investigated the effect of allelic polymorphisms in MHC-restricted peptides from AMA1 on *ex-vivo* malaria parasite-specific IFN-γ responses in subjects with a history of natural exposure to malaria infections using allelic AMA1 peptide sequences from the 3D7, 7G8, FVO, FC27 and tm284 *P. falciparum* strains. Our choice of AMA1 as an antigen for assessing allelic polymorphisms is on the basis that despite the demonstrated vaccine potential of AMA1, immune responses elicited by this polymorphic antigen have mostly been effective against parasites expressing the vaccine variant AMA1 antigen but poorly effective against parasites that express other AMA1 variants [8,9].

## Materials and methods

### Ethics

This study protocol was approved by the Institutional Review Board of the Noguchi Memorial Institute for Medical Research (NMIMR). The NMIMR IRB has a US Government Federal-wide Assurance (FWAA00001824) from the Office for Human Research Protections. Written informed consent was sought from all seven study subjects who willingly agreed to participate in the study and met the inclusion criteria.

### Study site, participants and sample collection

This study was carried out within the Legon community where the University of Ghana is located. Legon is approximately 10 Km northeast of the Accra city centre in the Greater Accra Region of Ghana, and malaria transmission is very low and limited to periods of rainfall between March and November. Study subjects were recruited at the end of March 2017. Study subjects were all males between the ages of 24 and 60 years and resident in the study area. Eligibility criteria for the study were as follows; participation in a previous study by the same group [23], in which participants' HLA A and B types were determined, age above 18 years, male or non-pregnant female or a nursing mother, haemoglobin >10 g/dl, and absence of known immunodeficiency (>400 CD4$^+$ T cells/μl). Seven subjects who participated in the said previous study, were still resident in the study area and met the other inclusion criteria were convenience sampled. All seven subjects tested negative for malaria parasites by light microscopy.

Sixty ml (60 ml) of venous blood was collected per subject into heparinized tubes. PBMCs were isolated from blood within four hours of collection by a gradient centrifugation method. Briefly, blood diluted in an equal volume of RPMI 1640 supplemented with L-glutamine and penicillin-streptomycin was slowly layered on Ficoll-Paque medium in 50 ml Falcon tubes. The tubes were centrifuged at 694 x g for 10 minutes to separate PBMCs from packed red cells at the bottom of the tubes and from diluted plasma as the top layer. The PBMC layer was pipetted into another Falcon tube and the cells washed with 5% Foetal Calf Serum in RPMI 1640 and spun at 391 x g for 7 minutes. After one more wash cycle, cells were counted and rested in an incubator at 37 ˚C, 5% CO$_2$ for at least 6 h before use in *ex vivo* ELISpot assays.

### Predicted 3D7 class 1 epitopes and matching variant sequences

For each of the seven study subjects, the known HLA class 1 A and B allele types were used to predict their recognition of 9-10mer peptides from the *P. falciparum* AMA1 antigen (3D7 strain, GenBank accession number U65407) using the NetMHC algorithm [24]. Peptides that have a predicted $IC_{50}$ of 500 nm or less were deemed as strong binders and therefore selected. The full-length sequence of the 3D7 strain *P. falciparum* AMA1 antigen was aligned with the corresponding AMA1 sequences from the 7G8 (GenBank accession number EU586371), FVO (GenBank accession number AY588147), FC27 (GenBank accession number M27133) and tm284 (GenBank accession number EU586507) parasite strains using a Microsoft Excel-based alignment algorithm for identification of sequence regions that show amino acid variability. 9-10mer 3D7 AMA1 peptides that were predicted to bind with high affinity to subject HLA alleles but have amino acid variability were selected.

### Peptide synthesis

The 3D7 peptides along with corresponding variant AMA1 peptides from the other parasite strains were synthesized commercially (Alpha Diagnostics Inc, San Antonio, TX, USA) based on the FMOC solid phase peptide synthesis technology [25]. All synthesized peptides had > 91% purity and were originally lyophilized. Peptides were initially dissolved in DMSO and diluted with sterile plain RPMI-1640 to a stock concentration of 10 mg/ml. Before use, the stock peptides were further diluted to a working concentration of 20 μg/ml in RPMI-1640 containing 1% penicillin–streptomycin, 1% L-glutamine and 10% normal human AB serum.

### ELISpot assay

IFN-γ ELISpot assays were carried out as previously described [23]. Briefly, multiscreen plates (Millipore Corporation, USA) were coated with 100 μl/well of 15 μg/ml monkey anti-human IFN-γ monoclonal antibodies (Mabtech AB, USA) in 0.1 M bicarbonate buffer, pH 9.6, and incubated overnight at 4˚C. RPMI-1640 was used to wash the plates six times, after which plates were blocked for at least 2 h with 10% human serum in RPMI-1640. Freshly isolated and rested PBMCs (400,000 cells/well) from each subject were added to triplicate wells with all the subject-specific allelic peptides to yield a final concentration of 10 μg/ml. Concanavalin A (Con A, Sigma Aldrich, USA) at 0.625 μg/ml and CEF (Cellular Technology Ltd, USA) at 2.0 μg/ml were added as positive controls to plates for all assays. Subject PBMCs incubated with medium only were used as negative controls. After PBMC incubation for 36 h at 37˚C, 5% $CO_2$, plates were washed six times with 250 μl/well of wash buffer (PBS containing 0.05% Tween 20) and incubated with 100 μl/well of 1 μg/ml biotinylated anti-IFN-γ monoclonal antibody (Mabtech, USA) diluted in 0.5% fetal calf serum (FCS) in PBS for 3 h at room temperature. Plates were afterwards washed six times and incubated with 100 μl/well of 1 μg/ml alkaline-phosphatase-conjugated streptavidin (Mabtech, USA) for 1 h at room temperature. Plates were again washed six times with 250 μl/well of wash buffer and then three times with plain PBS before incubation with an enzyme-specific chromogenic substrate (Bio-Rad, USA) for 15 min at room temperature. After the substrate incubation period, plates were washed under tap water and air-dried overnight at room temperature. An automated ELISpot plate reader (AID GmbH, Germany) was then used to determine the number of IFN-γ-producing cells in the form of spots per well. The data obtained was exported into Microsoft Excel worksheet for further analysis.

## Statistical and data analysis

For all controls and stimulants, the number of spots in replicate wells was averaged and ELISpot responses were calculated as the spot forming cells per million (sfc/m) PBMCs by multiplying by a factor of 2.5. Any replicate that contributed more than 50% of the variation for any triplicate set was excluded from the analysis; this was observed in 13% of all stimulations. For any stimulant, a positive IFN-γ response (POS) was defined as having at least a doubling of sfc/m in test wells relative to control wells with unstimulated cells, in addition to a difference of at least ten sfc/m between test and control wells, otherwise, the result was considered negative (NEG) [23]. For peptides within an allelic set, sfc/m were compared by either Student $t$ test (for paired peptides) or by One-way ANOVA (for more than 2 peptides) as appropriate with the GraphPad Prism 8.0 (GraphPad Software Inc, San Diego, CA, USA) in an attempt to assess the effect of polymorphism on peptide allele-specific responses. Differences in sfc/m between peptides within an allele set were considered statistically significant at $p < 0.05$.

## Results

### Variability in AMA1 alleles

Alignment of the full-length sequence of the 3D7 strain *P. falciparum* AMA1 antigen with the corresponding allelic AMA1 sequences from the 7G8, FVO, FC27 and tm284 parasite strains is presented in Fig 1. Amongst the five AMA1 peptide alleles compared, a total of 40 polymorphic residue positions, representing 6.4% of all residues, were identified. The 3D7 and FC27 AMA1 alleles had the least number of nine polymorphic positions between them, 3D7 and 7G8 AMA1 had 28 polymorphic positions between them while the 3D7 and FVO alleles had the highest number of 30 polymorphic positions between them (Fig 1). A total of 42 3D7 strain AMA1 peptides with at least one polymorphic site were bioinformatically predicted as potential HLA A- and B-binding epitopes, and following the alignment with the other four allelic AMA1 sequences, an additional 49 peptides with variability in at least one amino acid residue position relative to the predicted 3D7 consensus peptides were identified. Allelic peptides from the different parasite strains were grouped together as an allelic set, and details of all 91 peptides, grouped into 42 allelic sets, are presented in S1 Table. Thirty-five (35) of the 42 predicted 3D7 peptides were shared with one or more of the 4 other parasite strains, while the seven remaining peptides were unique to the 3D7 clone only.

### Peptide-specific ELISpot responses

Seven adult subjects were recruited for this study and all tested negative for malaria parasites by microscopy at the time of sample collection. Freshly isolated PBMCs from all subjects were stimulated with allelic AMA1 peptides and the positive control stimulants Con A (for cell viability) and CEF (for ability to respond to a CD8 peptide pool in responders). All seven study subjects tested positive for both Con A and CEF, except for subject 4 who tested negative for CEF. The 91 HLA class 1-restricted peptides were grouped into 42 allelic sets which covered A01, A02, A03, B07, B27 and B58 supertypes. Seven of the 42 allele sets had three variant peptides each and the remaining 35 allelic sets had two variant peptides (S1 Table). Thirteen of the 42 allelic sets were tested against PBMCs from multiple subjects, based on their predicted binding to the subjects' HLA types. The least number of allelic sets tested against a single subject was three and the highest number of allelic sets tested was 16 (S1 Table).

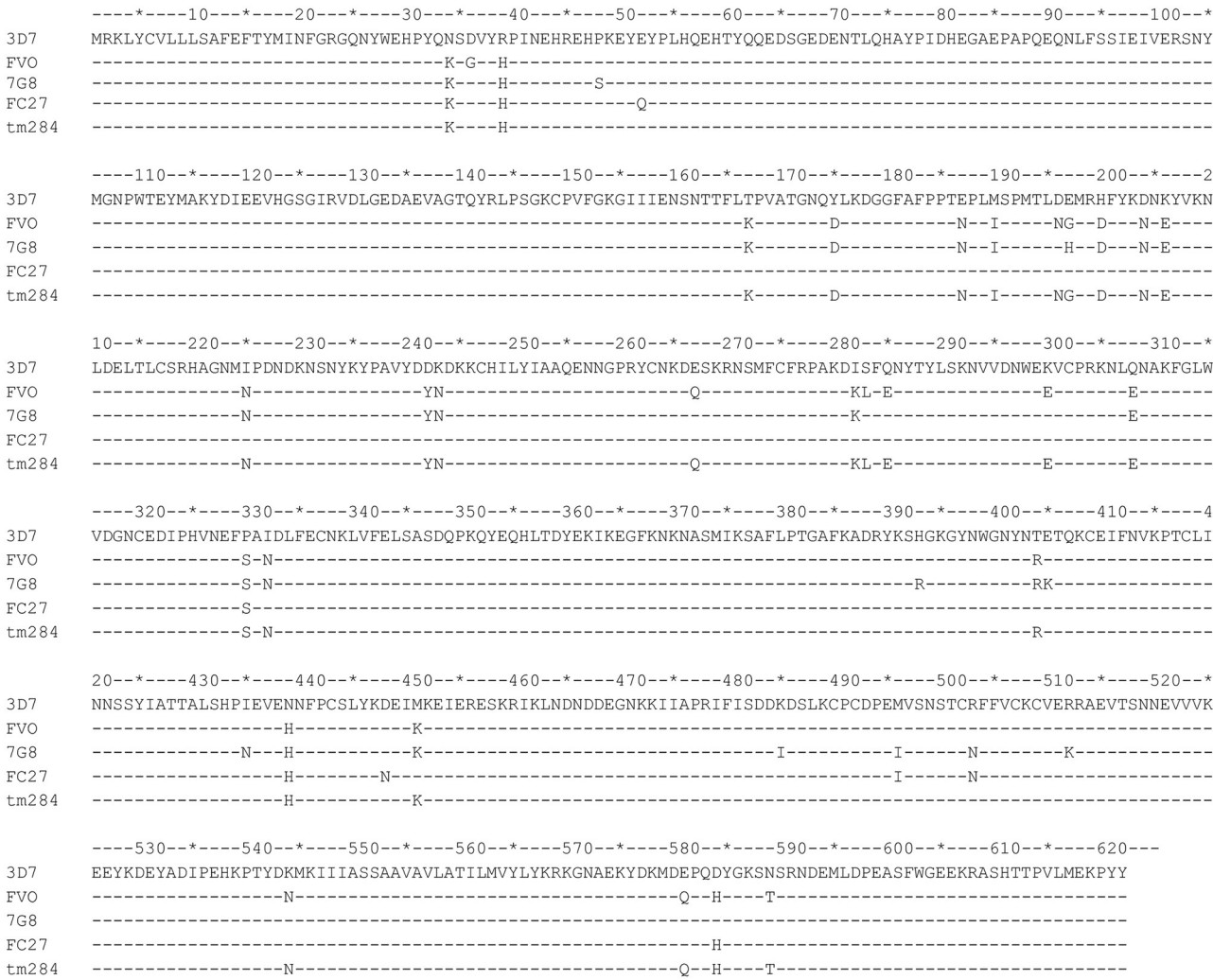

**Fig 1. Alignment of *P. falciparum* AMA1 sequences from the five parasite strains.** Sequences for the 3D7, FVO, 7G8, FC27 and tm284 parasite strains were sourced from GenBank and aligned using a Microsoft Excel-based alignment algorithm to identify polymorphic residues across the different variants.

### Subject v01 (HLA A01A03, A03, B07, B44)

A total of 34 peptides, representing 16 allelic sets, were tested against PBMCs from subject v01. The average responses to PBMC stimulations by all peptides ranged from 0 to 66 sfc/m. Of the 16 allelic sets tested (S1 Table), six had one peptide per allelic set testing positive, while all peptides within two allelic sets tested positive. Thus, ten peptides in eight allelic sets tested positive against PBMCs from subject v01 (Table 1). Aside comparing peptides for immune response positivity, we also performed a statistical comparison of peptide responses within allele sets to determine if differences in amino acids affected the stimulatory effect of peptides. Out of these eight allelic sets, responses (sfc/m) to peptides within four allelic sets (50%) were significantly different from responses to their corresponding allelic peptide(s) ($P < 0.05$) (Table 1). Also, for the two allele sets that had all component peptides testing positive, magnitude of responses to the two peptides in one set were not significantly different (IEVENNFPC and IEVE**H**NFPC, $P = 0.41$), while for the other positive set, responses to the

**Table 1. Outcome of subject v01 PBMC stimulation with allelic peptide sets.**

| HLA Supertype | HLA Allele | Parasite strains | Peptides | Ave sfc/m PBMCs | Results | P Value |
|---|---|---|---|---|---|---|
| A03 | A*68:01 | 7G8/FVO/tm284 | AVYD**YN**DKK | 18 | **POS** | |
| | | 3D7/FC27 | AVYDDKDKK | 0 | NEG | |
| A01A03 | A*30:01 | 7G8/FVO/tm284 | CSRHAGNM**N** | 9 | NEG | 0.49 |
| | | 3D7/FC27 | CSRHAGNMI | 13 | **POS** | |
| A03 | A*68:01 | FVO/tm284 | D**KLFE**NYTY | 5 | NEG | **0.0002** |
| | | 7G8 | D**KS**FQNYTY | 43 | **POS** | |
| | | 3D7/FC27 | DISFQNYTY | 0 | NEG | |
| A03 | A*68:01 | 7G8 | FVCKCVE**K**R | 32 | **POS** | **0.012** |
| | | 3D7/FVO/FC27/tm284 | FVCKCVERR | 6 | NEG | |
| B44 | B*18:01 | FVO/FC27/tm284 | IEVE**H**NFPC | 12 | **POS** | 0.41 |
| | | 3D7 | IEVENNFPC | 10 | **POS** | |
| A03 | A*68:01 | FVO/tm284 | **I**SPMTL**NG**MR | 13 | **POS** | 0.2 |
| | | 7G8 | **I**SPMTLD**H**MR | 6 | NEG | |
| | | 3D7/FC27 | MSPMTLDEMR | 6 | NEG | |
| B44 | B*18:01 | 7G8/FVO/tm284 | NEF**S**A**N**DLF | 14 | **POS** | **0.004** |
| | | 3D7 | NEFPAIDLF | 50 | **POS** | |
| A03 | A*68:01 | 7G8/FVO/tm284/FC27 | **H**NFPCSLYK | 3 | NEG | **0.0004** |
| | | 3D7 | NNFPCSLYK | 66 | **POS** | |

Data presented as sfc/m PBMCs (rounded to nearest whole number), after subtraction of background (unstimulated cell) sfc/m responses. For this subject, the average positive control (Con A) response was 164 sfc/m PBMCs (after background subtraction), and the average background response was 5 sfc/m PBMCs.

Data presented is for eight allelic sets where at least one peptide within a set tested positive.

Amino acids highlighted red and underlined are the variant residues relative to the 3D7 reference allele peptide.

POS is a positive ELISpot response and NEG is a negative response, based on positivity criteria described in Materials & Methods. Statistical significance was estimated either by the student t test for allele sets with two peptides or by ANOVA for allele sets with three peptides.

3D7 strain peptide (NEFPAIDLF) were significantly higher than responses to the corresponding allele in the other strains (NEF**S**A**N**DLF, P = 0.004). Forty percent (40%) of the ten peptides that elicited positive responses were of the 3D7 parasite strain type while the remaining 60% were non-3D7 strain peptides.

We further assessed the peptide induced IFN-γ response patterns in relation to the HLA predictions for the 3D7 reference peptide. The predicted A03 3D7 epitope AVYDDKDKK, conserved for the FC27 strain, tested negative while the variable sequence AVYD**YN**DKK, conserved amongst the 7G8/FVO/tm284 strains gave a positive response. Further, the predicted 3D7 A03 epitope, DISFQNYTY, conserved for the FC27 and the variable FVO/tm284 peptide DKLFENYTY both gave negative responses, while the variable 7G8 sequence, DKSFQNYTY, gave a positive response. Also, the predicted A03 3D7 epitope FVCKCVERR, conserved for FVO/FC27/tm284 strains tested negative while the variable sequence 7G8 FVCKCVE**K**R gave a positive response. The predicted B44 3D7 epitope NEFPAIDLF tested positive and the variant sequence NEF**S**A**N**DLF, conserved for the 7G8, FVO, FC27 and tm284 strains also tested positive but with a comparatively lower magnitude compared to the 3D7 variant sequence response. Finally, the predicted 3D7 A03 epitope, NNFPCSLYK gave a strong positive response while the, variable 7G8/FVO/tm284/FC27 peptide HNFPCSLYK was negative.

**Table 2. Outcome of subject v02 PBMC stimulation with allelic peptide sets.**

| HLA Supertype | HLA Allele | Parasite strains | Peptides | Ave sfc/m PBMCs | Results | P Value |
|---|---|---|---|---|---|---|
| A01A03 | A*30:01 | 7G8/FVO/tm284 | CSRHAGNM**N** | 14 | **POS** | 0.35 |
| | | 3D7/FC27 | CSRHAGNMI | 11 | **POS** | |
| A02 | A*68:02 | 7G8 | **K**TQKCEIFNV | 9 | NEG | **0.006** |
| | | 3D7/FVO/FC27/tm284 | ETQKCEIFNV | 28 | **POS** | |
| A01A03 | A*30:01 | 7G8 | RYKS**R**GKGY | 18 | **POS** | 0.64 |
| | | 3D7/FVO/FC27/tm284 | RYKSHGKGY | 21 | **POS** | |

Data presented as sfc/m PBMCs, after subtraction of background (unstimulated cell) sfc/m responses. For this subject, the average positive control (Con A) response was 288 sfc/m PBMCs (after background subtraction), and the average background response was 5 sfc/m PBMCs.

Data presented is for three allelic sets where at least one peptide within a set tested positive.

Amino acids highlighted red and underlined are the variant residues relative to the 3D7 reference allele peptide.

POS is a positive ELISpot response and NEG is a negative response, based on positivity criteria described in Materials & Methods. P values obtained after comparison of responses against the two or three peptides within the allelic set.

## Subject v02 (HLA A01A03, A02, B07, B44)

Six single peptides representing three allelic peptide sets were tested against PBMCs from subject v02 (Table 2). The average responses to PBMC stimulations by all peptides ranged from 9 to 28 sfc/m. All peptides in two allelic sets tested positive and there were no statistically significant differences between the positive responses in each of these two allele sets. For the third allele set, only one of the two component peptides tested positive, and the difference in sfc/m between these two peptides was statistically significant (P = 0.006, Table 2). Thus, five peptides in three allelic sets elicited positive responses against PBMCs from subject v02. Sixty percent (60%) of the five peptides that elicited positive responses were of the 3D7 parasite strain type while the remaining 40% were non-3D7 strain peptides. The predicted 3D7 epitope ETQK-CEIFNV for HLA supertype A02, conserved for the FVO/FC27/tm284 strains gave a positive response while the variant sequence KTQKCEIFNV in the 7G8 strain was negative. Also, the two predicted HLA A01A03 3D7 epitopes both gave similar positive responses as the other variant sequence against PBMCs from this subject.

## Subject v03 (HLA A01, A02, B07, B44)

A total of 29 peptides, sub-grouped into 13 allelic sets, were tested against PBMCs from subject v03. Responses to PBMC stimulations by these peptides ranged from 0 to 268 average sfc/m. Out of the 14 allelic sets, two had all component peptides eliciting positive responses and there were no significant differences in the sfc/m values within each set (Table 3). A third allelic set had a single peptide eliciting a positive response, and this positive response was statistically significantly higher than the corresponding allele response (Table 3). Overall, five peptides in three allelic sets elicited positive responses against PBMCs from subject v03. Sixty percent (60%) of the five peptides that elicited positive responses were of the 3D7 parasite strain type while the remaining 40% were non-3D7 strain peptides.

## Subject v04 (HLA A01, A03, B27, B27)

A total of 20 peptides, sub-grouped into nine allelic sets were tested against PBMCs from subject v04. Responses to PBMC stimulations by these peptides ranged from 0 to 27 average sfc/m. One allelic set had the 3D7 allele peptide being positive and the response was marginally significantly higher than that of the corresponding allele peptide (P = 0.04, Table 4). Thus, one

**Table 3. Outcome of subject v03 PBMC stimulation with allelic peptide sets.**

| HLA Supertype | HLA allele | Parasite strains | Peptides | Ave sfc/m PBMCs | Results | P Value |
|---|---|---|---|---|---|---|
| A01 | A*30:02 | FVO/tm284 | D**KLFE**NYTY | 20 | NEG | 0.13 |
| | | 7G8 | D**K**SFQNYTY | 33 | NEG | |
| | | 3D7/FC27 | DISFQNYTY | 40 | **POS** | |
| B07 | B*35:01 | 7G8/FC27 | **I**VSNSTC**N**F | 268 | **POS** | **0.003** |
| | | 3D7/FVO/tm284 | MVSNSTCRF | 79 | **POS** | |
| B44 | B*44:03 | FVO/7G8/tm284 | NEF**SAN**DLF | 163 | **POS** | 0.55 |
| | | 3D7 | NEFPAIDLF | 148 | **POS** | |

Data presented as sfc/m PBMCs, after subtraction of background (unstimulated cell) sfc/m responses. For this subject, the average positive control (Con A) response was 361 sfc/m PBMCs (after background subtraction), and the average background response was 47 sfc/m PBMCs.

Data presented is for three allelic sets where at least one peptide within a set tested positive.

Amino acids highlighted red and underlined are the variant residues relative to the 3D7 reference allele peptide.

POS is a positive ELISpot response and NEG is a negative response, based on positivity criteria described in Materials & Methods. P values obtained after comparison of responses against the two or three peptides within the allelic set.

peptide in a single allelic set elicited a positive response against PBMCs from subject v04, and this peptide was of the 3D7 strain parasite. The predicted 3D7 epitope EHREHPKEY for HLA B27, conserved for the FVO/FC27/tm284 strains gave a positive response while the variant sequence EHREH**S**KEY in the 7G8 strain was negative.

## Subjects v05, v06 and v07

PBMCs from subjects v05 (HLA A03, A03, B27, B58), v06 (HLA A02, A02, B07, B44) and v07 (A01, A03, B07, B58) did not yield any positive response against any peptide stimulants. Nine single peptides, sub-grouped into four allelic sets were tested against PBMCs from subject v05. Responses to PBMC stimulations by these peptides ranged from 0 to 10 average sfc/m. For subject v06, nine peptides, sub-grouped into four allelic sets were tested. Responses to PBMC stimulations by these peptides were all 0 average sfc/m. Finally, 22 peptides sub-grouped into 10 allelic sets were tested against PBMCs from subject v07. Responses to PBMC stimulations by these peptides ranged from 0 to 10 average sfc/m.

Overall, four of the seven (57%) study subjects had at least a single positive peptide response. A total of 18 peptides, present in 12 of the 42 allelic sets (28.6%), elicited positive responses in these four subjects. Of the 12 allelic sets that yielded positive responses, four sets had all component peptides testing positive against some subject PBMCs, while a single

**Table 4. Outcome of subject v04 PBMC stimulation with allelic peptide sets.**

| HLA Supertype | HLA Allele | Parasite strains | Peptides | Ave sfc/m PBMCs | Results | P Value |
|---|---|---|---|---|---|---|
| B27 | B*15:03 | 7G8 | EHREH**S**KEY | 11 | NEG | **0.04** |
| | | 3D7/FVO/FC27/tm284 | EHREHPKEY | 27 | **POS** | |

Data presented as sfc/m PBMCs, after subtraction of background (unstimulated cell) sfc/m responses. For this subject, the average positive control (Con A) response was 524 sfc/m PBMCs (after background subtraction), and the background response was 14 sfc/m PBMCs.

Data presented is for a single allelic set where one peptide tested positive.

Amino acids highlighted red and underlined are the variant residues relative to the 3D7 reference allele peptide.

POS is a positive ELISpot response and NEG is a negative response, based on positivity criteria described in Materials & Methods. P values obtained after comparison of magnitude of responses against the two or three peptides within the allelic set.

peptide in each of eight allelic sets tested positive. The two peptides in one allele set (NEF-**S**A**N**DLF and NEF**PA**IDLF) both tested positive against PBMCs from two different subjects (v01 and v03), both of whom express an HLA B44 supertype. Also, the 3D7 peptide DIS-FQNYTY, presented by an HLA A01 supertype, tested positive against PBMCs from subject v03 (Table 3), while the corresponding variant 7G8 peptide D<u>K</u>SFQNYTY, possibly presented by an HLA A03 supertype, tested positive with PBMCs from subject v01 (Table 1). Among all the 18 peptides that tested positive, nine (50%) were from the *P. falciparum* strain 3D7 and the other nine were from the other 4 strains including 7G8, FVO, FC27, and tm284. Of the 18 positive peptides, three yielded very potent IFN-γ responses (> 100 sfc/m PBMCs), all against PBMCs from subject v03, six gave moderately potent responses (between 30 and 100 sfc/m) and the remaining nine were responses between 10 and 30 sfc/m PBMCs.

## Effect of amino acid position and substitution on peptide immunogenicity

T cell activation requires multiple interactions between MHC molecules, the pathogen peptide being presented and the T cell receptor. We therefore hypothesized that amino acid substitutions in the pathogen peptide could influence MHC and TCR binding and hence the induction of the appropriate T cell response. In order to investigate this, we compared differences in amino acid residue position and substitutions within the 12 allele sets that had at least one peptide being positive.

Six peptides that elicited positive responses in three subjects (v01, v02 and v04) showed statistically significant differences from their corresponding allelic peptides that did not elicit positive IFN-γ T cell responses. Additionally, all peptides in two allele sets (NEFPAIDLF and NEFSANDLF for subject v01, MVSNSTCRF and IVSNSTCNF for subject v03), despite eliciting positive responses, yielded sfc/m values that were statistically significantly different (Tables 1 and 3). Thus, at least eight allelic peptide sets had differences in amino acid composition or position of change that could result in one peptide within the set having statistically significantly higher magnitude of the elicited T cell response compared to the other(s).

Of the eight positive peptides whose corresponding alleles showed statistically significantly lower magnitude of responses, a total of 12 amino acid substitutions were present. Three substitutions occurred at each of amino acid positions 1 and 6. Substitutions at position 1 were from a polar to a positively charged residue (Table 1), from a negatively charged to a positively charged residue (Table 2) and from one non-polar residue to another (Table 3). At position 6, substitutions were from polar to a positively charged residue (Table 1) and two from non-polar to a polar residue (Tables 1 and 4). Two substitutions occurred at position 5 and they both involved changes from polar to negatively charged residues. In addition, there were single amino acid substitutions occurring at all other amino acid residue positions except positions 7 and 9. For allele sets with significant differences in magnitude of the sfc/m response amongst the component alleles, there was a higher frequency of substitutions in residue positions 1 and 6 (Fig 2). Overall, 75% (9/12) of the substitutions resulted in a change in polarity, while the remaining 25% (3/12) of substitutions resulted in no change in amino acid polarity.

In contrast, for the 6 allelic sets with no statistically significant differences in responses, a total of 11 amino acid substitutions were observed. Three amino acid substitutions occurred at position 5 and involved amino acids changes from a positively charged to a non-polar residue (Table 1), one positively charged residue to another (Table 2) and a polar to a negatively charged residue (Table 3). There were also single amino acid substitutions occurring at each of the other 8 amino acid positions; three of these substitutions were from neutral-to-neutral residues, another three were from neutral to polar residues and the remaining two were from neutral to polar residues. Thus, 36% (4/11) of the amino acid substitutions resulted in no change

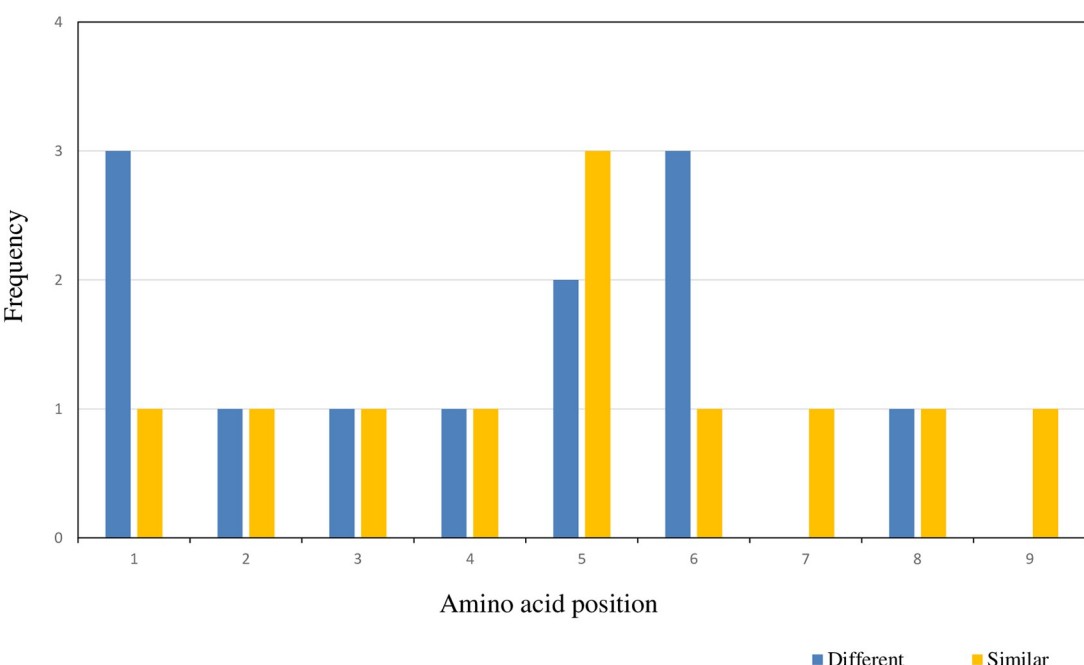

**Fig 2. Frequency of polymorphic residue distribution at each amino acid position within test peptides.** Numbers are based on data presented in Tables 1 to 4. For each residue position, amino acid substitutions that resulted in statistically significant differences in the magnitude of IFN-γ responses between allelic peptides are designated as "Different" (in blue colour) and substitutions that did not result in statistically significant changes in IFN-γ responses between allelic peptides are designated "Similar" (in yellow colour).

in the polarity of the substituted amino acid, while 64% (7/11) of the substitutions resulted in a change in polarity. While there were no amino acid substitutions at positions 7 and 9 for allelic sets with statistically significantly different responses, amino acid substitutions were found at all residue positions in peptides belonging to allelic sets with significantly similar responses.

## Discussion

Polymorphisms in several malaria parasite antigens have been shown to significantly impact the functional quality of antibody responses and hence the protective efficacy of vaccines based on these targets [7–9]. There are indications that this may also be the case even for T vaccines such as attenuated whole parasite and gene-based vaccines that are designed to elicit T cell mediated immunity, where protection in experimental challenge studies is highly strain-dependent [10,26]. Thus, antigen polymorphisms may play a critical role in the induction of the right quality and magnitude of both antibody and T cell responses. The purpose of this study was therefore to determine the effect of polymorphisms in peptide sequences on the induction of IFN-γ responses by identifying immunodominant peptides from the 3D7 parasite strain AMA1 and comparing IFN-γ responses of these with corresponding responses from allelic peptides from other parasite strains in subjects naturally exposed to malaria parasites in an endemic area.

Overall, four of the seven study subjects elicited positive responses to 18 peptides within 12 of the 42 different allelic sets tested. Fifty percent (50%) of positive peptides were from the *P. falciparum* strain 3D7, with the other 50% coming from 4 other strains, and this is a strong indication of the possibility of infection of study subjects by multiple parasite variants, a phenomenon that has been reported [27–29]. All seven study subjects were parasite-negative by

microscopy at the time of sampling, and we did not assess infecting parasite diversity by PCR. The absence of microscopic parasites however affirms our earlier reports [23,30] that natural malaria infections can elicit significant T cell responses, even in the absence of significant active malaria parasite infections.

Three of the 18 positive peptides had IFN-γ responses above 100 sfc/m PBMCs while those of the other 15 positive peptides were between 10 and 100 sfc/m PBMCs. Usually, ELISpot IFN-γ responses in PBMCs from naturally infected persons are lower than those in PBMCs from vaccinated malaria-naïve subjects. We have however shown in a previous study that it is still possible under natural infection conditions to identify immunodominant peptides with the generally lower responses [30]. In that study, we identified four immunodominant peptides from the CSP parasite antigen in a naturally exposed population, and these peptides had previously been described in vaccinated malaria-naïve persons with IFN-γ responses that were at least 3-fold higher [31,32] than our measured responses [30]. We have also shown in another previous study [23] that our positivity criteria are robust enough and confirmed the positivity of 82% of immunodominant peptides that were identified using more stringent response positivity definition that is applied to vaccine-induced ELISpot IFN-γ responses [32,33]. Three of the seven subjects generally had IFN-γ responses below 10 sfc/m PBMCs over background responses, and this might have been due to the very high background responses in these subjects.

The 3D7 peptide DISFQNYTY, presented by an HLA A01 allele (A*30:02) was positive against PBMCs from subject v03 (Table 3), while the corresponding variant 7G8 peptide DKSFQNYTY, possibly presented by an HLA A03 allele (A*68:01), tested positive with PBMCs from subject v01 (Table 1). Also, the allelic peptides NEFPAIDLF (3D7)/NEFSANDLF (other strains) tested positive against PBMCs from both subject v01 and v03, albeit at significantly different response magnitudes. These allelic peptides were most likely presented by the HLA allele B*18:01 in subject v01 but by the HLA allele B*44:03 in subject v03. Both of these HLA allele types belong to the HLA B44 supertype, and these observations collectively indicate promiscuous binding of the peptides to HLA molecules, a phenomenon that has been previously described [34,35]. These observations also point to the parasite variants that different subjects have been exposed to, and validates the notion that subjects will respond to predicted epitopes matching their HLA based on what they were primed with through infection or vaccination.

We further investigated the effect of amino acid substitutions within allelic peptides on the induction of positive IFN-γ responses. Eight of 12 allele sets had at least one peptide that elicited a statistically significant IFN-γ response (P < 0.05) compared with the other allele(s) within those respective sets. The predominant positions where changes that accounted for different responses occurred were at residues 1 and 6. Of the 12 amino acid differences identified in these positive peptides, 75% of substitutions resulted in a change in residue polarity while the remaining 25% of substitutions did not result in polarity changes. This is in comparison with 64% of substitutions that resulted in polarity changes in allelic peptide sets that elicited statistically similar levels of IFN-γ responses. While there seems to be no direct association between amino acid polarity at specific positions and the induction of IFN-γ responses, other factors such as the size and secondary structure of substituted amino acids [36] may also contribute to explaining these observations. The diversity in parasites that subjects have been exposed to could also be partly responsible for our observations, since different subjects made responses to the alternate forms of peptides within the same sets. These complex interactions are further complicated by the requirement of binding sites for both HLA molecules and T cell receptors on the peptide since specific anchor residues on both the MHC and TCR interact with corresponding residues in the peptides [14,15,37]. Amino acid substitutions within

peptides therefore affect peptide recognition by both the TCR and HLA molecules. These complex interactions may be best delineated using bioinformatics approaches, which are beyond the scope of the current experimental analysis presented in this paper.

Overall, while some amino acid substitutions abrogated the IFN-γ induction potential of immunodominant peptides or resulted in positive responses in both allele peptides albeit at different response magnitudes in some subjects, similar substitutions in peptides did not affect response induction in other subjects. Deciphering these has significant implications for the design of subunit T cell epitope-based vaccine expected to work through the induction of potent IFN-γ responses [33,38], since only a careful selection of relevant parasite strain-transcending peptides will ensure broad applicability of such vaccines. It is possible that a clearer observation can be made with analysis involving a much greater number of subjects. Other limitations of this study include the lack of molecular data for assessing sub-microscopic parasitaemia and our inability to concurrently assess the effect of peptide polymorphisms on peptide-MHC recognition by the TCR.

In summary, we have identified at least 18 peptides from variable regions of the multi-stage malaria parasite antigen AMA1 that have the capacity to elicit significant immune responses when used to stimulate PBMCs from naturally infected persons. Although we demonstrate an effect of some peptide residue substitutions on the induction of T cell responses, there is no clear pattern regarding which residue changes significantly influence T cell recognition since other similar substitutions did not yield the same effects. This immune modulatory effect could also in part be due to the effect of peptides polymorphisms on their recognition by the TCR, in addition to the MHC-mediated binding effect.

## Supporting information

**S1 Table. The 91 peptides tested against subject PBMCs grouped into 42 allelic sets.** For each peptide set, the table also shows the HLA alleles that were predicted to bind to the 3D7 variant peptides and the specific subjects whose PBMCs were tested.
(XLSX)

## Acknowledgments

We are grateful to technical staff of the Immunology Department of Noguchi Memorial Institute for Medical Research for assistance with PBMC cryopreservation and support for performance of assays. We are also grateful to study subjects for taking time to participate in this study.

**Disclaimer:** The views expressed in this article reflect the results of research conducted by the authors and do not necessarily represent the official policy or position of the Department of the Navy, Department of Defense nor the U.S. government.

## Author Contributions

**Conceptualization:** Omarine N. Nlinwe, Bjoern Peters, Eileen Villasante, Martha Sedegah, Kwadwo Asamoah Kusi.

**Data curation:** Omarine N. Nlinwe, Ebenezer A. Ofori, Kwadwo Akyea-Mensah, Eric Kyei-Baafour, Harini Ganeshan, Maria Belmonte, Martha Sedegah, Kwadwo Asamoah Kusi.

**Formal analysis:** Omarine N. Nlinwe, Kwadwo Asamoah Kusi.

**Funding acquisition:** Omarine N. Nlinwe.

**Investigation:** Omarine N. Nlinwe, Ebenezer A. Ofori, Kwadwo Akyea-Mensah, Eric Kyei-Baafour, Kwadwo Asamoah Kusi.

**Methodology:** Omarine N. Nlinwe, Ebenezer A. Ofori, Kwadwo Akyea-Mensah, Eric Kyei-Baafour, Harini Ganeshan, Maria Belmonte.

**Project administration:** Omarine N. Nlinwe, Kwadwo Asamoah Kusi.

**Resources:** Harini Ganeshan, Maria Belmonte, Bjoern Peters, Eileen Villasante, Martha Sedegah, Kwadwo Asamoah Kusi.

**Software:** Bjoern Peters.

**Supervision:** Martha Sedegah, Kwadwo Asamoah Kusi.

**Validation:** Omarine N. Nlinwe, Ebenezer A. Ofori, Kwadwo Akyea-Mensah, Eileen Villasante.

**Visualization:** Omarine N. Nlinwe, Eric Kyei-Baafour.

**Writing – original draft:** Omarine N. Nlinwe, Martha Sedegah, Kwadwo Asamoah Kusi.

**Writing – review & editing:** Ebenezer A. Ofori, Kwadwo Akyea-Mensah, Eric Kyei-Baafour, Harini Ganeshan, Maria Belmonte, Bjoern Peters, Eileen Villasante, Martha Sedegah, Kwadwo Asamoah Kusi.

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
