## [Decision Letter · Decision Letter 0]

23 Jun 2021

PONE-D-21-18237

Comparative analysis of the ex vivo IFN-Gamma Responses to CD8+ T cell epitopes within allelic forms of PfAMA1 in a Malaria Exposed Population

PLOS ONE

Dear Dr. Kusi,

Thank you for submitting your manuscript to PLoS ONE. After careful consideration, we felt that your manuscript requires substantial revision, following which it can possibly be reconsidered, thus governing the decision of a “major revision”. Although your manuscript was of interest to the reviewers, major concerns were related to study design and data interpretation.  According to the reviewer, the weakness of the manuscript was the small sample size and inferences based on such sample size is statistically challenging. In addition, there are several issues/concerns that should be addressed to the authors. 

For your guidance, a copy of the reviewers' comments was included below.

We look forward to receiving your revised manuscript.

Kind regards,

Luzia Helena Carvalho, Ph.D.

Academic Editor

PLOS ONE

Journal Requirements:

Reviewers' comments:

Reviewer's Responses to Questions

**Comments to the Author**

1. Is the manuscript technically sound, and do the data support the conclusions?

Reviewer #1: Yes

2. Has the statistical analysis been performed appropriately and rigorously? 

Reviewer #1: Yes

3. Have the authors made all data underlying the findings in their manuscript fully available?

Reviewer #1: Yes

4. Is the manuscript presented in an intelligible fashion and written in standard English?

Reviewer #1: Yes

5. Review Comments to the Author

Reviewer #1: The study attempt to understand allelic polymorphisms in the malarial antigen apical membrane antigen 1 (AMA1) and the effect on ex vivo T cell-specific IFN-γ responses in a population with lifelong exposure to malaria.

The manuscript is well written.

The strength of the manuscript is that it speaks to the design of malaria vaccine candidate, a global need and very relevant to today’s malaria science.

The weakness of the manuscript is that the results are based on ONLY 7 candidate/individuals study. This sample size is very small and inferences based on such sample size is statistically challenging.

These said, there are a number of issues/concerns that I believe would further improve the manuscript if looked into.

6. PLOS authors have the option to publish the peer review history of their article (what does this mean?). If published, this will include your full peer review and any attached files.

Reviewer #1: **Yes: **Moses Okpeku

---

## [Author Response · Author response to Decision Letter 0]

22 Jul 2021

A point by point response to the reviewer comments have been uploaded along with the revised manuscript files

---

## [Decision Letter · Decision Letter 1]

16 Aug 2021

PONE-D-21-18237R1

Comparative analysis of the ex vivo IFN-Gamma Responses to CD8+ T cell epitopes within allelic forms of PfAMA1 in a Malaria Exposed Population

PLOS ONE

Dear Dr. Kusi,

Thank you for submitting your manuscript to PLoS ONE. After careful consideration, we felt that your manuscript still requires substantial revision, following which it can possibly be reconsidered, thus governing the decision of a “major revision”. According to the reviewer # 2, Major concern was related to the statistical methods. Also the raw data (spots/well or at least sfc/m ± SD) should be included as supplementary information. For your guidance, a copy of the reviewer comments was included below. We therefore invite you to submit a revised version of the manuscript paying close attention to the specific points raised by the reviewer.   

We look forward to receiving your revised manuscript.

Kind regards,

Luzia Helena Carvalho, Ph.D.

Academic Editor

PLOS ONE

Reviewers' comments:

Reviewer's Responses to Questions

**Comments to the Author**

1. If the authors have adequately addressed your comments raised in a previous round of review and you feel that this manuscript is now acceptable for publication, you may indicate that here to bypass the “Comments to the Author” section, enter your conflict of interest statement in the “Confidential to Editor” section, and submit your "Accept" recommendation.

Reviewer #1: All comments have been addressed

Reviewer #2: (No Response)

2. Is the manuscript technically sound, and do the data support the conclusions?

Reviewer #1: Yes

Reviewer #2: Yes

3. Has the statistical analysis been performed appropriately and rigorously? 

Reviewer #1: Yes

Reviewer #2: I Don't Know

4. Have the authors made all data underlying the findings in their manuscript fully available?

Reviewer #1: Yes

Reviewer #2: No

5. Is the manuscript presented in an intelligible fashion and written in standard English?

Reviewer #1: Yes

Reviewer #2: Yes

6. Review Comments to the Author

Reviewer #1: (No Response)

Reviewer #2: In this study, the authors analyse the effect of polymorphisms affecting CD8+ T cell epitopes from PfAMA-1 sequences on IFNgamma responses measured on PBMCc from 7 subjects living in malaria endemic regions. Those polymorphisms represent PfAMA-1 allelic sequences from 3D7, 7G8, FVO, FC27 and tm284 parasite strains. The authors state that the aminoacidic substitutions do not show a pattern of association with response abrogation.

Main comments:

-Although the authors explain the limitations of their conclusions regarding specifically the (very) low number of subjects included in their study, they extrapolate anyway their findings to a “malaria exposed population” in the title, abstract (line 30) and introduction (line 102). In the opinion of this reviewer, the less assertive term used in the discussion “subjects naturally exposed to malaria” (line 438) is more in agreement with the study, as 7 subjects are hardly representative of a population.

-The main concern I found is related to statistical and data analysis mentioned in Materials and Methods section (lines 189-201) and the data showed in the Tables 1-4. The authors state a positivity criteria for IFNgamma response as “at least a doubling of sfc/m in test wells relative to controls (…) in addition to a difference of at least ten spots between test and control wells”. Ten spots in wells containing 400,000 cells/well correspond to 25 sfc/m (million); in that case, 25 sfc/m should be the minimum to consider the sample positive, in the case of a background of zero spots. As in the tables 1-4, there are samples considered positive with as low as 10 sfc/m, either the positivity criteria are wrongly stated, or the analysis is incorrect, or something is missing. From the line 459 in Discussion “responses below 10 sfc/m PBMCs over background responses” I gather that the sfc/m described in the tables are the values after subtraction of background. If that if the case, this is not clear for the reader and should be stated in the MM section and in the legends of the tables. Also, it is not clear if these criteria are applied to each replicate or to averages, please clarify.

It is also confusing that the averages of sfc/m, after all calculations (average of triplicates and extrapolation from 400,000 cells to 1,000,000) are mentioned as whole numbers in the tables. This apparently substantial approximation should be mentioned.

It should be interesting to know if the stimulations which needed to exclude one replicate (13%) corresponded to one particular subject, as this could be related or not to the high background mentioned for some subjects.

Taking into account all these concerns, in my opinion the raw data of spots/well or at least sfc/m ± SD corresponding to the samples mentioned in the tables (both control and stimulated) should be presented in a supplementary table as supporting information.

7. PLOS authors have the option to publish the peer review history of their article (what does this mean?). If published, this will include your full peer review and any attached files.

Reviewer #1: **Yes: **Moses Okpeku

Reviewer #2: No

---

## [Author Response · Author response to Decision Letter 1]

18 Aug 2021

Dear Editor,

We are grateful once again for the reviewer’s comments which will ultimately help improve upon the manuscript. Kindly find our point-by-point responses to the comments below.

Reviewer #2: In this study, the authors analyse the effect of polymorphisms affecting CD8+ T cell epitopes from PfAMA-1 sequences on IFNgamma responses measured on PBMCc from 7 subjects living in malaria endemic regions. Those polymorphisms represent PfAMA-1 allelic sequences from 3D7, 7G8, FVO, FC27 and tm284 parasite strains. The authors state that the aminoacidic substitutions do not show a pattern of association with response abrogation.

Main comments:

-Although the authors explain the limitations of their conclusions regarding specifically the (very) low number of subjects included in their study, they extrapolate anyway their findings to a “malaria exposed population” in the title, abstract (line 30) and introduction (line 102). In the opinion of this reviewer, the less assertive term used in the discussion “subjects naturally exposed to malaria” (line 438) is more in agreement with the study, as 7 subjects are hardly representative of a population.

Response

We agree with the reviewer on this point, and have accordingly revised the title to read as follows:

Comparative analysis of the ex vivo IFN-Gamma Responses to CD8+ T cell epitopes within allelic forms of PfAMA1 in Subjects with Natural Exposure to Malaria

We have also edited the relevant portions of the manuscript to reflect this

Comment

-The main concern I found is related to statistical and data analysis mentioned in Materials and Methods section (lines 189-201) and the data showed in the Tables 1-4. The authors state a positivity criteria for IFNgamma response as “at least a doubling of sfc/m in test wells relative to controls (…) in addition to a difference of at least ten spots between test and control wells”. Ten spots in wells containing 400,000 cells/well correspond to 25 sfc/m (million); in that case, 25 sfc/m should be the minimum to consider the sample positive, in the case of a background of zero spots. As in the tables 1-4, there are samples considered positive with as low as 10 sfc/m, either the positivity criteria are wrongly stated, or the analysis is incorrect, or something is missing. From the line 459 in Discussion “responses below 10 sfc/m PBMCs over background responses” I gather that the sfc/m described in the tables are the values after subtraction of background. If that if the case, this is not clear for the reader and should be stated in the MM section and in the legends of the tables. Also, it is not clear if these criteria are applied to each replicate or to averages, please clarify.

Response

The positivity criteria as defined is based on the spot forming cells per million (sfc/m) PBMCs and not on the 400,000 PBMCs used in the assays. The sfc/m value is considered standardized for across study comparisons, since other studies could plate different numbers of PBMCs. These same criteria and approach have been applied in our earlier studies [1–3]. We have however further clarified this in the data analysis section of the manuscript. 

The data presented in Tables 1 to 4 are indeed the sfc/m after background subtraction. All reads, including positive controls, test stimulants and background, were first converted to sfc/m before background subtraction for test stimulants and positive controls. These were stated in the statistical analysis section and have been better clarified now. We have also included additional footnotes to Tables 1 to 4 to make these clearer as requested.

Comment

It is also confusing that the averages of sfc/m, after all calculations (average of triplicates and extrapolation from 400,000 cells to 1,000,000) are mentioned as whole numbers in the tables. This apparently substantial approximation should be mentioned.

Response

This was deliberate as we are dealing with cells and cannot be counting fractions of cells. The numbers have been rounded off to the nearest whole number, and we do not believe that this should have a significant effect on the data as presented. We have however indicated that these numbers are rounded to the nearest whole number in the table footnote, at least for Table 1.

Comment

It should be interesting to know if the stimulations which needed to exclude one replicate (13%) corresponded to one particular subject, as this could be related or not to the high background mentioned for some subjects.

Response

These are single replicate spot readings that looked highly different from the remaining two duplicate readings in 13 % of stimulations, with no more than two such cases per subject. It is therefore not related to observations for any particular subject.

COmment

Taking into account all these concerns, in my opinion the raw data of spots/well or at least sfc/m ± SD corresponding to the samples mentioned in the tables (both control and stimulated) should be presented in a supplementary table as supporting information.

Response

We have presented the sfc/m values (after background subtraction) as well as the unstimulated background responses for each subject as a footnote to their respective data tables, so this should allow readers to estimate the sfc/m before background subtraction.

Once again, we are grateful to the reviewer for pointing out these deficiencies and to the editor for allowing us to address these concerns and re-submit our manuscript.

References

1. Anum D, Kusi KA, Ganeshan H, Hollingdale MR, Ofori MF, Koram KA, et al. Measuring naturally acquired ex vivo IFN-γ responses to Plasmodium falciparum cell-traversal protein for ookinetes and sporozoites (CelTOS) in Ghanaian adults. Malaria Journal. 2015;14:20.

2. Ganeshan H, Kusi KA, Anum D, Hollingdale MR, Peters B, Kim Y, et al. Measurement of ex vivo ELISpot interferon-gamma recall responses to Plasmodium falciparum AMA1 and CSP in Ghanaian adults with natural exposure to malaria. MalarJ. 2016;15:55.

3. Kusi KA, Aggor FE, Amoah LE, Anum D, Nartey Y, Amoako-Sakyi D, et al. Identification of Plasmodium falciparum circumsporozoite protein-specific CD8+ T cell epitopes in a malaria exposed population. PLOS ONE. 2020;15:e0228177.

---

## [Decision Letter · Decision Letter 2]

26 Aug 2021

Comparative analysis of the ex vivo IFN-Gamma Responses to CD8+ T cell epitopes within allelic forms of PfAMA1 in Subjects with Natural Exposure to Malaria

PONE-D-21-18237R2

Dear Dr.  Kusi,

We’re pleased to inform you that your manuscript has been judged scientifically suitable for publication and will be formally accepted for publication once it meets all outstanding technical requirements.

Kind regards,

Luzia Helena Carvalho, Ph.D.

Academic Editor

PLOS ONE

Additional Editor Comments (optional):

Reviewers' comments:

Reviewer's Responses to Questions

**Comments to the Author**

1. If the authors have adequately addressed your comments raised in a previous round of review and you feel that this manuscript is now acceptable for publication, you may indicate that here to bypass the “Comments to the Author” section, enter your conflict of interest statement in the “Confidential to Editor” section, and submit your "Accept" recommendation.

Reviewer #2: All comments have been addressed

2. Is the manuscript technically sound, and do the data support the conclusions?

Reviewer #2: Yes

3. Has the statistical analysis been performed appropriately and rigorously? 

Reviewer #2: Yes

4. Have the authors made all data underlying the findings in their manuscript fully available?

Reviewer #2: Yes

5. Is the manuscript presented in an intelligible fashion and written in standard English?

Reviewer #2: Yes

6. Review Comments to the Author

Reviewer #2: Although the authors did not upload raw data as supplementary information, the data included in the data analysis section, and mainly in the footnotes of Tables 1-4, are enough to answer my concerns about the statistical methods.

7. PLOS authors have the option to publish the peer review history of their article (what does this mean?). If published, this will include your full peer review and any attached files.

Reviewer #2: No

---

## [Editor Report · Acceptance letter]

31 Aug 2021

PONE-D-21-18237R2 

Comparative analysis of the *ex vivo* IFN-Gamma Responses to CD8+ T cell epitopes within allelic forms of P*f*AMA1 in Subjects with Natural Exposure to Malaria 

Dear Dr. Kusi:

I'm pleased to inform you that your manuscript has been deemed suitable for publication in PLOS ONE. Congratulations! Your manuscript is now with our production department. 

Kind regards, 

on behalf of

Dr. Luzia Helena Carvalho 

Academic Editor

PLOS ONE